# Expression Profiles of Circulating MicroRNAs in XELOX-Chemotherapy-Induced Peripheral Neuropathy in Patients with Advanced Gastric Cancer

**DOI:** 10.3390/ijms23116041

**Published:** 2022-05-27

**Authors:** Yeongdon Ju, Young Mi Seol, Jungho Kim, Hyunwoo Jin, Go-Eun Choi, Aelee Jang

**Affiliations:** 1Department of Clinical Laboratory Science, College of Health Sciences, Catholic University of Pusan, Busan 46252, Korea; lrdrlr@naver.com (Y.J.); jutosa70@cup.ac.kr (J.K.); jjinhw@cup.ac.kr (H.J.); 2Clinical Trial Specialist Program for In Vitro Diagnostics, Brain Busan 21 Plus Program, Graduate School, Catholic University of Pusan, Busan 46252, Korea; 3Division of Hematology-Oncology, Department of Internal Medicine and Biomedical Research Institute, Pusan National University Hospital, Busan 49241, Korea; seol2100@hanmail.net; 4Department of Nursing, University of Ulsan, Ulsan 44610, Korea

**Keywords:** gastric cancer, chemotherapy-induced peripheral neuropathy, microRNAs, bioinformatics analysis

## Abstract

Gastric cancer (GC) is one of the most common cancers and a leading cause of cancer deaths around the world. Chemotherapy is one of the most effective treatments for cancer patients, and has remarkably enhanced survival rates. However, it has many side effects. Recently, microRNAs (miRNAs) have been intensively studied as potential biomarkers for cancer diagnosis and treatment monitoring. However, definitive biomarkers in chemotherapy-induced peripheral neuropathy (CIPN) are still lacking. The aim of this study was to identify the factors significant for neurological adverse events in GC patients receiving XELOX (oxaliplatin and capecitabine) chemotherapy. The results show that XELOX chemotherapy induces changes in the expression of hsa-miR-200c-3p, hsa-miR-885-5p, and hsa-miR-378f. Validation by qRT-PCR demonstrated that hsa-miR-378f was significantly downregulated in CIPN. Hsa-miR-378f was identified as showing a statistically significant correlation in GC patients receiving XELOX chemotherapy according to the analysis of differentially expressed (DE) miRNAs. Furthermore, 34 potential target genes were predicted using a web-based database for miRNA target prognostication and functional annotations. The identified genes are related to the peptidyl-serine phosphorylation and regulation of alternative mRNA splicing with enrichment in the gastric cancer, neurotrophin, MAPK, and AMPK signaling pathways. Collectively, these results provide information useful for developing promising strategies for the treatment of XELOX-chemotherapy-induced peripheral neuropathy.

## 1. Introduction

Gastric cancer (GC) is one of the most serious health problems, being the fifth most common malignancy globally, and remains the third most common cause of cancer death worldwide [1,2]. According to GLOBOCAN 2020 data, GC was estimated to have caused the death of approximately 800,000 people worldwide [2,3]. Chemotherapy is a widely considered major treatment method for advanced GC [4]. However, patients undergoing chemotherapy may experience side effects, such as retching, vomiting, diarrhea, and alopecia, because chemotherapy kills rapidly growing normal cells [5]. XELOX chemotherapy consists of capecitabine and oxaliplatin [6]. Although oxaliplatin is not nephrotoxic or ototoxic, it causes numerous side effects, the major one being neurotoxicity. In 70% of cases, prolonged treatment with oxaliplatin causes critical chronic peripheral neuropathy, requiring the patient to stop chemotherapy [7]. Therefore, it is necessary to develop strategies to confirm the clinical prognosis for neuropathy in GC patients receiving chemotherapy.

MicroRNAs (miRNAs) are small single-stranded RNAs of about 19 to 24 nucleotides [8]. They regulate the expression of their target messenger RNAs (mRNAs) [9] and can promote tumor growth [10], infiltration [11], angiogenesis [12], and antigenic escape [13]. Research has shown that miRNAs may serve as either oncogenes or tumor suppressors in most types of cancer, including gastric [14], pancreatic [15], and colorectal cancers [16]. Multiple previous studies have demonstrated an miRNA–mRNA network in GC [17,18]. However, to the best of our knowledge, there is no systematic and inclusive analysis of miRNA–mRNA interaction networks associated with XELOX-chemotherapy-induced peripheral neuropathy (CIPN) in GC patients.

Combining data-mining techniques with bioinformatics analysis can provide researchers with an unprecedented advantage in discovering therapeutic targets and novel cancer biomarkers [19]. In this study, we aimed to uncover novel microRNA biomarkers of XELOX-chemotherapy-induced peripheral neuropathy in patients with advanced gastric cancer through the analysis of differentially expressed (DE) miRNAs and miRNA–mRNA interaction networks.

## 2. Results

### 2.1. Study Group Characteristics

Blood samples from a total of 32 GC patients who received at least two cycles of XELOX chemotherapy were divided into two groups. Group 1 comprised patients who received fewer than four cycles of XELOX chemotherapy, and group 2 comprised patients who received four or more cycles of XELOX chemotherapy. Table 1 presents the clinical and statistical characteristics of the patients. The mean age of group 1 was 55.64 ± 11.69 years, and that of group 2 was 55.44 ± 11.71 years for both males and females. All the patients had received pathological diagnoses of gastric cancer at stages IIA–IV according to the eighth edition of the AJCC tumor staging system. The CIPN symptoms, as determined by the National Cancer Institute Common Terminology Criteria for Adverse Events (NCI-CTCAE) grades, were greater than or equal to 1 in both groups. In samples of patients, Common Terminology Criteria for Adverse Events (CTCAE) Grade 2 (moderate symptoms) occurred more frequently in patients with four or more cycles of chemotherapy than in patients with fewer than four cycles of chemotherapy. Particularly, CTCAE Grade 3 (severe symptoms) was only observed in patients with four or more cycles of chemotherapy. Therefore, we established a group based on four cycles of chemotherapy.

### 2.2. Analysis of Small RNA Sequencing Data

Small RNA sequencing analysis was performed by ebiogen, Inc. (Seoul, South Korea). The DE miRNAs between patients who received fewer than four cycles of XELOX chemotherapy and those who received 4–7 cycles of XELOX chemotherapy were identified using miRNA expression profile data. The obtained data were analyzed using ExDEGA GraphicPlus version 2.0 (ebiogen). The miRNA expression profile was generated by volcano plot analysis using GraphPad Prism version 6.0 (Figure 1a). The miRNA expression profile data highlighted 2588 mature miRNAs; four of the miRNAs downregulated were differentially expressed between patients receiving fewer than 4 cycles and 4–7 cycles of XELOX chemotherapy. The four downregulated DE miRNAs were determined according to a threshold of *p* < 0.05 and fold change > 2. The four downregulated DE miRNAs identified were implemented as heatmaps and were plotted using MultiExperiment Viewer (MeV) version 4.9.0. (Figure 1b). These miRNAs included hsa-miR-885-5p, hsa-miR-378f, hsa-miR-200c-3p, and hsa-miR-4666a-3p. The basic information for the four DE miRNAs is listed in Table 2.

### 2.3. Validation of Differentially Expressed (DE) miRNAs by Quantitative Real-Time (RT) PCR Analysis

qRT-PCR analysis was used to validate the results from the RNA sequencing. The expression levels of the DE miRNAs were analyzed following the chemotherapy cycles. The expression levels of miR-378f, miR-200c-3p, and miR-885-5p were higher in the group receiving chemotherapy for fewer than 4 cycles than in the group receiving chemotherapy for 4–7 cycles. The expression of miR-4666a-3p was not detectable in the qRT-PCR measurement. The expression of miR-378f was significantly downregulated (*p* = 0.0041) (Figure 2).

### 2.4. Correlation Analysis of Differentially Expressed (DE) miRNAs in GC Patients Receiving XELOX Chemotherapy

Correlation analysis was performed to identify the relationship between DE miRNAs and GC patients receiving XELOX chemotherapy. We found that only miR-378f, among the three DE miRNAs, was correlated with GC patients receiving XELOX chemotherapy. By using Pearson’s chi-squared tests, the differential expression of miR-378f in GC patients receiving XELOX chemotherapy was determined to be statistically significant (Table 3).

### 2.5. Target Prediction Analysis for miRNAs

An exact prediction of miRNA targets is important for specificizing the functions of DE miRNAs. As shown in Figure 3, a total of 34 predicted target genes of hsa-miR-378f (blue) were obtained. The DE miRNA and predicted target genes are noted in Appendix A.

### 2.6. Gene Ontology (GO) Annotation Analysis

Gene ontology (GO) analysis was used for identifying and visualizing the biological pathways related to the target genes. The GO terms for the target genes are presented in Figure 4. These target genes are associated with miR-378f. For the GO term biological process (BP), the analysis showed peptidyl-serine phosphorylation (GO:0018105), regulation of alternative mRNA splicing via spliceosome (GO:0000381), and peptidyl-threonine phosphorylation (GO:0018107). The significantly enriched cellular component (CC) terms included midbody (GO:0030496), cytosol (GO:0005829), and nucleus (GO:0005634). The molecular function (MF) terms included identical protein binding (GO:0042802), protein kinase C binding (GO:0005080), and RNA binding (GO:0003723).

### 2.7. Kyoto Encyclopedia of Genes and Genomes (KEGG) Pathway Enrichment Analysis

A pathway enrichment analysis of target genes indicated that they are involved in gastric cancer, neurotrophin signaling, mitogen-activated protein kinase (MAPK) signaling, adenosine monophosphate-activated protein kinase (AMPK) signaling, forkhead box protein O (FoxO) signaling, repressor activator protein 1 (Rap1) signaling, cellular senescence, and apoptosis (Figure 5 and Table 4). The predicted target genes are represented in Appendix A.

## 3. Discussion

The number of cancer survivors is anticipated to increase to 22.1 million by 2030, as the effectiveness of antineoplastic regimens has successfully extended life expectancy [20]. As a result, the cancer survival rates for patients are also rising, and the incidence of chemotherapy-induced peripheral neuropathy (CIPN) may increase in the future [21]. CIPN can cause troubles in normal daily life that compromise health-related quality of life [22,23]. Moreover, despite the constant development of symptom-related strategies for treating CIPN, preventive and curative interventions remain difficult [24]. Therefore, a more relevant approach for CIPN using circulating biomarkers is needed. Recent studies have suggested that specific microRNAs (miRNAs) can regulate various genes and pathways involved in neuropathic pain [25,26,27]. In this study, we conducted research to identify the miRNAs in the plasma that were differentially expressed (DE) between gastric cancer (GC) patients receiving fewer than 4 cycles of XELOX (capecitabine and oxaliplatin) chemotherapy and those receiving 4–7 cycles of XELOX chemotherapy.

First, we carried out small RNA sequencing analysis and identified four miRNAs remarkably downregulated in GC patients receiving 4–7 cycles of chemotherapy compared with GC patients receiving fewer than 4 cycles: miR-4666a-3p, miR-885-5p, miR-200c-3p, and miR-378f. To validate our results obtained from the RNA sequencing analysis, the expression of the four miRNAs was compared by qRT-PCR with quantification using 5S ribosomal RNA as control. The expression of hsa-miR-378f was remarkably downregulated compared with that in the control group, while hsa-miR-4666a-3p was not detectable by qRT-PCR. We also found that only miR-378f was associated with GC patients receiving XELOX chemotherapy.

miRNAs are involved in the regulation of protein and gene expression and may also be involved in pain processing pathways [28]. In a rat model study of oxaliplatin-induced peripheral neuropathy, miR-30b-5p attenuated neuropathic pain via voltage-gated sodium channels [29]. The expression of miR-141-5p was found to be downregulated by oxaliplatin in a neuropathic pain rat model [30]. In addition, miR-15b mediated chronic neuropathic pain induced by oxaliplatin [31]. However, in our study of GC patients with CIPN, we observed no difference in the expression levels of these miRNAs between the analyzed groups. On the other hand, it has been reported that miR-378 reduces neuropathic aches by negatively targeting enhancer of zeste homolog 2 (EZH2), and the miR-378/EZH2 axis plays an important role in the development of neuropathic pain [32]. Moreover, it has been reported that miR-378 is downregulated in rats with chronic constriction injury, and the overexpression of miR-378 reduces pain [33]. Therefore, it is rational to deduce that miR-378 may act on neuropathic pain.

To further investigate, we utilized bioinformatics analyses and performed Kyoto Encyclopedia of Genes and Genomes (KEGG) pathway and gene ontology (GO) function annotation analysis. Intriguingly, our results show that the apoptotic process may be associated with one of the most noticeable differences in CIPN. Apoptosis plays significant roles in cancer, the immune system, and the response to chemotherapy [34,35,36]. Additionally, apoptosis participates in nerve injury and the formation of neuropathic pain [37,38]. However, the specific mechanism of the apoptosis in CIPN ought to be investigated in the future. The adenosine monophosphate-activated protein kinase (AMPK) signaling pathways are significant in the pathogenesis of neuropathic pain [39]. Moreover, mitogen-activated protein kinase (MAPK) signaling pathways are implicated in the mechanism of neuropathic pain in rats with chronic compression of the dorsal root ganglion [40]. Generally, our KEGG pathway analysis demonstrated that the abnormal activities of these pathways during neuropathy can be triggered by our selected miRNAs.

Other studies have concentrated on circulating miRNAs as biomarkers for vincristine- and paclitaxel-induced neuropathic pain [41,42]. However, our study concentrated on GC patients receiving XELOX chemotherapy. Whether our results are apposite to GC patients receiving other chemotherapy agents, such as cisplatin and paclitaxel, remains to be determined by further research. Moreover, further research with larger sample sizes and biological experiments are needed to support and validate our findings.

## 4. Materials and Methods

### 4.1. Clinical Samples

From May 2020 to December 2021, blood was collected from 32 advanced GC patients undergoing XELOX chemotherapy (combined with oxaliplatin and capecitabine chemotherapy) at Pusan National University Hospital (PNUH). Whole blood was collected into a tube containing an RNA stabilizer. Then, immediately after blood collection, the plasma from each sample was separated and stored at −80 °C.

### 4.2. Grading of Chemotherapy-Induced Peripheral Neuropathy (CIPN)

The severity of the chemotherapy-induced peripheral neuropathy (CIPN) was clinically graded by an oncologist using the National Cancer Institute Common Terminology Criteria for Adverse Events (NCI-CTCAE) version 4.03. The CIPN grades in NCI-CTCAE include motor and sensory neuropathy grades, with each item scored from 1 (mild symptoms) to 5 (death) [43,44,45].

### 4.3. Analysis Pipeline for the Study

The workflow and analysis pipeline used in this study is shown in Figure 6. Small RNA sequencing was performed using the NextSeq500 system (Illumina, San Diego, CA, USA) by randomly selecting 2 patients from group 1 and 4 patients from group 2. Group 1 contained patients who received fewer than 4 cycles of XELOX chemotherapy, while group 2 contained patients who received 4 or more cycles of XELOX chemotherapy. We performed differentially expressed (DE) miRNA analysis to identify significant differences. Then, we conducted miRNA target prediction using miRNet. Additionally, to investigate the potential pathogenesis of XELOX-chemotherapy-induced peripheral neuropathy in advanced GC patients, gene ontology (GO) annotation analysis using the DAVID database and pathway enrichment analysis for target genes using the Kyoto Encyclopedia of Genes and Genomes (KEGG) database was performed. Finally, we validated the candidate miRNA expression by qRT-PCR using different patient samples classified into two groups.

### 4.4. RNA Isolation

Total RNA samples were obtained from the plasma using the TRIzol reagent (Invitrogen, CA, USA) following the manufacturer’s protocols. The RNA quality was assessed using an RNA 6000 Pico Chip (Agilent Technologies, Amstelveen, The Netherlands) and an Agilent 2100 Bioanalyzer system (Agilent Technologies, Amstelveen, the Netherlands). A NanoDrop 2000 spectrophotometer system (Thermo Fisher Scientific, Waltham, MA, USA) was used to quantify the RNA for conducting RNA-seq analysis.

### 4.5. Library Preparation and Sequencing

Libraries were constructed using a NEBNext Multiplex Small RNA Library Prep kit (New England Biolabs, MA, USA). The amplification step of the library construction was performed using polymerase chain reactions (PCRs), and the products were then purified using AMPure XP beads (Beckmancoulter, Pasadena, CA, USA) and a QIAquick PCR Purification Kit (Qiagen, Hilden, Germany). The Agilent 2100 Bioanalyzer and a high-sensitivity DNA kit (Agilent Technologies, Santa Clara, CA, USA) were used to assess the size distribution and yield of the small RNA libraries. High-throughput sequencing was performed using single-ended 75 bp sequences and the NextSeq500 system (Illumina, CA, USA).

### 4.6. Data Analysis

The BAM file (alignment file) was obtained by mapping the sequence reads using the Bowtie2 software tool. The mature miRNA sequence was used as a reference for mapping. The read counts mapped to mature miRNA sequences were extracted from the alignment file using Bioconductor, which uses R version 3.2.2, and bedtools version 2.25.0. The read counts of the annotated miRNAs were normalized using the trimmed mean of m-values (TMM) and counts per million (CPM) method for comparison between the samples.

### 4.7. miRNA Expression Profiles

A volcano plot and heatmap were used to identify DE miRNAs and determine the miRNA expression profile. Volcano plot analysis was performed using GraphPad Prism version 6.0, and heatmap analysis was performed using MultiExperiment Viewer version 4.9.0.

### 4.8. Quantitative Real-Time (RT)-PCR

qRT-PCR and cDNA synthesis were performed using a QuantStudio^TM^ 7 Flex Real-Time PCR system (Applied Biosystems, Foster City, CA, USA). After synthesizing cDNA using a miRCURY LNATM RT Kit (Qiagen Sciences, Germantown, MD, USA), quantitative PCR was performed with primers (Appendix A) and the following thermal cycling conditions: 95 ℃ for 2 min, followed by 40 cycles of 10 sec at 95 ℃ and 1 min at 56 ℃ for 50 cycles. Real-time PCR was carried out using a miRCURY LNATM SYBR Green PCR kit (Qiagen Sciences). As a control, 5S ribosomal RNA (5S rRNA) was used. The expression levels of the DE miRNAs were determined using the 2−ΔΔCt method (ΔΔCt = ΔCt [target-reference]_sample_ − ΔCt [target-reference]_mean of controls_) [46]. Each sample was processed in triplicate.

### 4.9. miRNA Target Prediction

Based on the results from the small RNA sequencing, miRNA–mRNA interactions were predicted using miRNet (https://www.mirnet.ca/ accessed on 3 May 2022), which is a miRNA target computational prediction tool. The database on miRNet is a powerful web-based tool that integrates data from two servers for experimentally verified miRNA gene targets, TarBase and miRTarBase [47].

### 4.10. Gene Ontology (GO) Annotation

Gene ontology (GO) annotations against the predicted target genes of DE miRNAs were performed. The genes were sorted using the DAVID database (https://david.ncifcrf.gov/ accessed on 3 May 2022), which is a tool for investigators to comprehend the biological mechanisms associated with large gene lists [48]. The GO analysis included biological process (BP), cellular component (CC), and molecular function (MF).

### 4.11. Kyoto Encyclopedia of Genes and Genomes (KEGG) Pathway Enrichment

Version 3.7.0 of the Cytoscape software, which is based on ClueGO, was used to visualize enriched pathways associated with the Kyoto Encyclopedia of Genes and Genomes (KEGG) database resource (kappa score = 0.4; *p*-value < 0.05).

### 4.12. Statistical Analysis

The statistical analysis was performed using the Statistical Product and Service Solutions (SPSS) version 26.0 (IBM, Armonk, NY, USA) and GraphPad Prism version 6.0 (GraphPad Software, San Diego, CA, USA) software. Differences between groups were determined using independent samples *t*-tests. The chi-squared test was performed to compare categorical variables. The tests were two-tailed, and *p*-values < 0.05 were considered to indicate statistically significant differences.

## 5. Conclusions

In summary, our study investigated the changes in miRNA expression occurring in response to XELOX-chemotherapy-induced peripheral neuropathy using small RNA sequencing. The results from multiple bioinformatics analyses indicate that only hsa-miR-378f could be involved in the processes of pathological change in XELOX-chemotherapy-induced peripheral neuropathy.

## Figures and Tables

**Figure 1 ijms-23-06041-f001:**
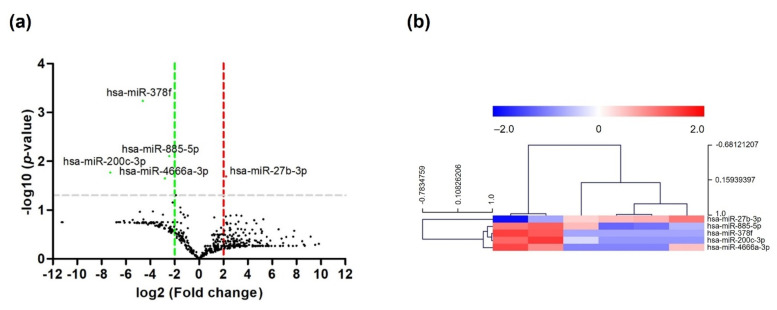
Volcano plot and heatmap for the identification of DE miRNAs. (**a**) Volcano plot of DE miRNAs in plasma samples of GC patients who received fewer than 4 cycles and received 4–7 cycles of XELOX chemotherapy. Log2 fold changes between patients who received fewer than 4 cycles and received 4–7 cycles of XELOX chemotherapy are plotted on the *x*-axis, and the –log10 of the *p*-value is plotted on the *y*-axis. Red: upregulated miRNA; green: downregulated miRNA; (**b**) heatmap of miRNA expression data from plasma samples of GC patients who received fewer than 4 cycles and received 4–7 cycles of XELOX chemotherapy; red: upregulated miRNA; blue: downregulated miRNA.

**Figure 2 ijms-23-06041-f002:**
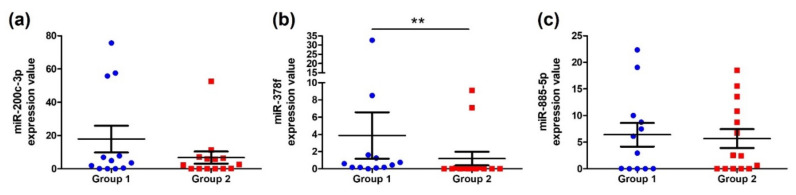
The expression levels of three DE miRNAs in groups 1 and 2: (**a**) miR-200c-3p, (**b**) miR-378f, (**c**) miR-885-5p. Group 1: received fewer than 4 cycles of XELOX chemotherapy. Group 2: received 4–7 cycles of XELOX chemotherapy. ** represents *p* < 0.01.

**Figure 3 ijms-23-06041-f003:**
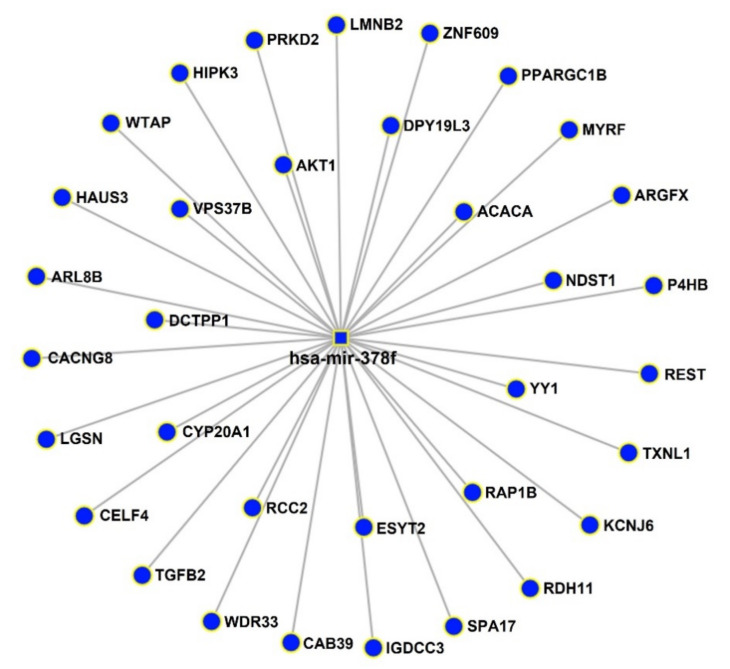
The interaction network of DE miRNA and their experimentally validated target genes. The network was generated using the miRNet tool.

**Figure 4 ijms-23-06041-f004:**
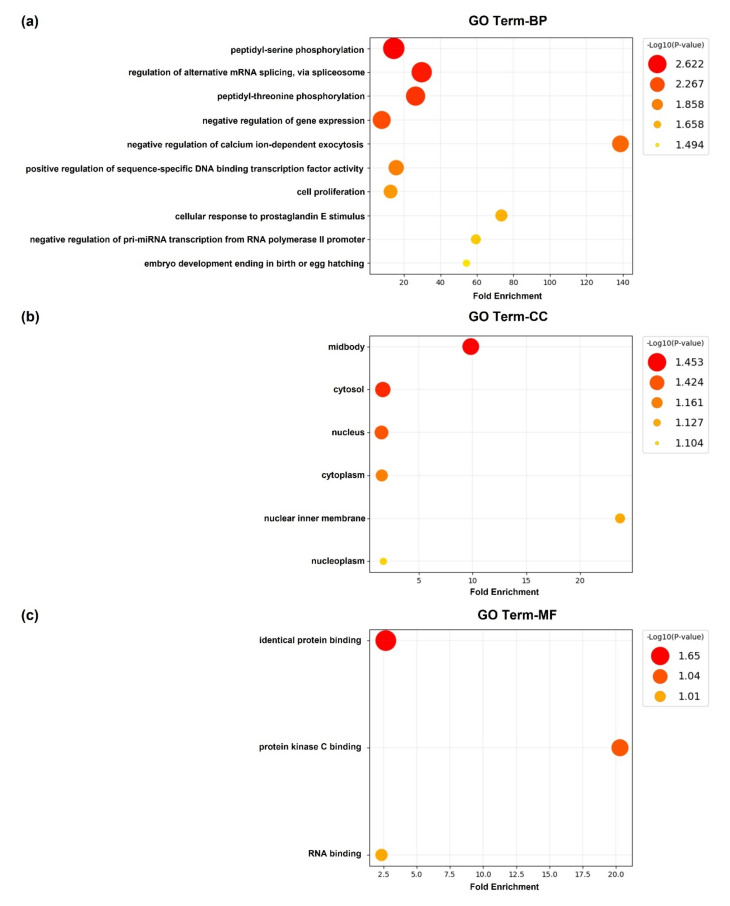
The gene ontology (GO) annotation analyses of the predicted target genes of the DE miRNAs. The top 10 most remarkably affected GO terms are listed along the *y*-axes. The *x*-axes denote fold enrichment. *p* < 0.05 was considered to indicate significant enrichment. The color and size of each dot denote –log10 (*p*-value). (**a**) The enrichment according to the biological process for the target genes of downregulated DE miRNAs, (**b**) the enrichment according to the cellular component for the target genes of downregulated DE miRNAs, (**c**) the enrichment according to the molecular function for the target genes of downregulated DE miRNAs.

**Figure 5 ijms-23-06041-f005:**
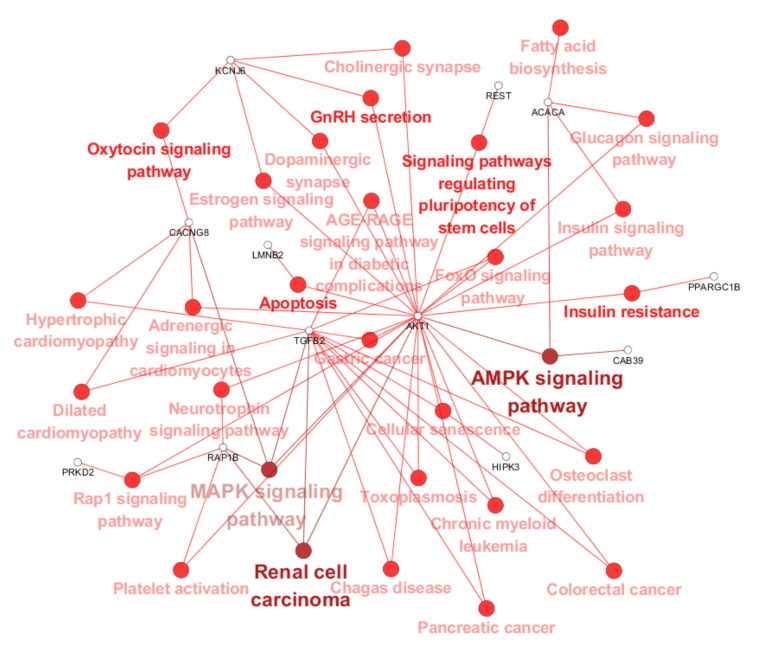
The interaction network represents KEGG pathways and potential target genes. Enriched pathways were obtained from the Kyoto Encyclopedia of Genes and Genomes (KEGG) database.

**Figure 6 ijms-23-06041-f006:**
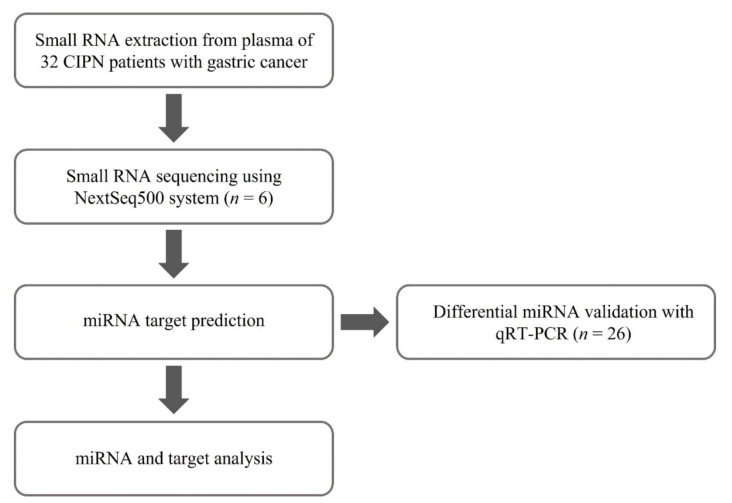
Overview of the analysis pipeline in this study.

**Table 1 ijms-23-06041-t001:** Clinicopathological features of GC patients with XELOX chemotherapy.

Variable	Group 1 (*n* = 14)	Group 2 (*n* = 18)	*p*-Value
**Gender, *n* (%)**			0.123
Male	9 (64.3)	16 (88.9)	
Female	5 (35.7)	2 (11.1)	
Age (years)			0.962
Mean ± SD	55.64±11.69	55.44±11.71	
**Chemotherapy cycles, *n* (%)**			<0.001
Cycle 2	4 (28.6)		
Cycle 3	10 (71.4)		
Cycle 4		5 (27.8)	
Cycle 5		6 (33.3)	
Cycle 6		4 (22.2)	
Cycle 7		3 (16.7)	
**Grade (NCI-CTCAE), *n* (%)**			<0.001
Grade 1	9 (64.3)	1 (5.6)	
Grade 2	5 (35.7)	11 (61.1)	
Grade 3		6 (33.3)	
**Primary tumor (T), *n* (%)**			0.704
T1	1 (7.1)	1 (5.6)	
T3	4 (28.6)	6 (33.3)	
T4a	7 (50.0)	10 (55.6)	
T4b	2 (14.3)	1 (5.6)	
**Regional lymph node (N), *n* (%)**			0.917
N0	1 (7.1)		
N1	2 (14.3)	8 (44.4)	
N2	6 (42.9)	1 (5.6)	
N3a	2 (14.3)	4 (22.2)	
N3b	3 (21.4)	5 (27.8)	
**Distant metastasis (M), *n* (%)**			0.442
M0	11 (78.6)	16 (88.9)	
M1	3 (21.4)	2 (11.1)	
**Stage (8th AJCC), *n* (%)**			0.73
IIA	1 (7.1)	1 (5.6)	
IIB		4 (22.2)	
IIIA	5 (35.7)	3 (16.7)	
IIIB	2 (14.3)	4 (22.2)	
IIIC	3 (21.4)	4 (22.2)	
IV	3 (21.4)	2 (11.1)	

Data are expressed as either frequency with percentages or means ± standard deviations; group 1: fewer than 4 cycles of XELOX chemotherapy; group 2: 4–7 cycles of XELOX chemotherapy; NCI-CTCAE: National Cancer Institute Common Terminology Criteria for Adverse Events; AJCC: American Joint Committee on Cancer; *p*-values were obtained using the independent samples *t*-test.

**Table 2 ijms-23-06041-t002:** List of DE miRNAs in GC patients with XELOX chemotherapy.

Downregulated DE miRNA	Accession	Fold Change	*p*-Value
hsa-miR-378f	MI0016756	−4.620	0.0006 ***
hsa-miR-885-5p	MI0005560	−2.449	0.008 **
hsa-miR-200c-3p	MI0000650	−7.292	0.017 *
hsa-miR-4666a-3p	MI0017296	−2.805	0.022 *

Fold change > 2.0; *** represents *p* < 0.001; ** represents *p* < 0.01; * represents *p* < 0.05.

**Table 3 ijms-23-06041-t003:** Differential expression of three miRNAs in GC patients in groups 1 and 2.

miRNA	Group 1		Group 2		χ^2^	*p*-Value	OR
	+	–	+	–			
hsa-miR-378f	9	3	2	12	9.758	0.0018 **	0.05556 (0.007613 ± 0.4054)
hsa-miR-885-5p	7	5	8	6	0.003752	0.9512	0.9524 (0.1998 ± 4.540)
hsa-miR-200c-3p	9	3	8	6	0.9104	0.34	0.4444 (0.08267 ± 2.389)

Group 1: fewer than 4 cycles of XELOX chemotherapy. Group 2: 4–7 cycles of XELOX chemotherapy. OR: odds ratio. ** represents *p* < 0.01.

**Table 4 ijms-23-06041-t004:** KEGG pathway analysis of target genes in GC patients with XELOX chemotherapy.

KEGG ID	KEGG Term	Gene Count	*p*-Value
KEGG:04010	MAPK signaling pathway	4	0.004
KEGG:04152	AMPK signaling pathway	3	0.003
KEGG:04015	Rap1 signaling pathway	3	0.012
KEGG:04218	Cellular senescence	3	0.005
KEGG:05211	Renal cell carcinoma	3	<0.001
KEGG:04921	Oxytocin signaling pathway	2	0.05
KEGG:04931	Insulin resistance	2	0.026
KEGG:04210	Apoptosis	2	0.04
KEGG:04550	Signaling pathways regulating pluripotency of stem cells	2	0.044
KEGG:04910	Insulin signaling pathway	2	0.04
KEGG:04922	Glucagon signaling pathway	2	0.03
KEGG:04725	Cholinergic synapse	2	0.03
KEGG:04728	Dopaminergic synapse	2	0.038
KEGG:04915	Estrogen signaling pathway	2	0.041
KEGG:04929	GnRH secretion	2	0.01
KEGG:04068	FoxO signaling pathway	2	0.037
KEGG:04261	Adrenergic signaling in cardiomyocytes	2	0.047
KEGG:04380	Osteoclast differentiation	2	0.04
KEGG:04611	Platelet activation	2	0.034
KEGG:04722	Neurotrophin signaling pathway	2	0.031
KEGG:04933	AGE-RAGE signaling pathway in diabetic complications	2	0.023
KEGG:05142	Chagas disease	2	0.023
KEGG:05145	Toxoplasmosis	2	0.028
KEGG:05210	Colorectal cancer	2	0.017
KEGG:05212	Pancreatic cancer	2	0.013
KEGG:05220	Chronic myeloid leukemia	2	0.013
KEGG:05226	Gastric cancer	2	0.047
KEGG:05410	Hypertrophic cardiomyopathy	2	0.018
KEGG:05414	Dilated cardiomyopathy	2	0.02
KEGG:00061	Fatty acid biosynthesis	1	0.041

GO: gene ontology; KEGG: Kyoto Encyclopedia of Genes and Genomes.

## Data Availability

The data presented in this study are available in Appendix A.

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
