# Peer review of "Expression Profiles of Circulating MicroRNAs in XELOX-Chemotherapy-Induced Peripheral Neuropathy in Patients with Advanced Gastric Cancer"

_ijms, 2022, doi:10.3390/ijms23116041_

Round 1

Reviewer 1 Report

This study aimed at identifying miRNAs associated with neurological side effects in GC patients who were receiving XELOX treatment. The authors found that XELOX treatment causes alterations in the expression of hsa-miR-22 200c-3p, hsa-miR-885-5p, and hsa-miR-378f, according to the findings. However, they could only validate these results for hsa-miR-378f using qRT-PCR, which is also the only one with correlation to GC. The authors present the ethics approval and informed consent protocol. The methods are well described, besides the bioinformatic analysis and the authors clearly state the limitations of the study. I have some concerns with this study presented point by point below.

Major comments

  • If the authors could only validate one miRNA and this miRNA was also the one which correlated with GC, why do GO, KEGG and network analysis with targets of two other miRNAs that were not validate nor correlated with GC?
    1. Suggestion: Redo the analysis with the predicted targets of hsa-miR-378f only.
  • 89: “using a volcano plot and heatmap.” Please refer to the methods used and not their graphic representation.
  • Section 4.3: This section lacks detail. It is impossible to reproduce these results without knowing which packages and tools were used to perform the analysis.
  • L-323-326: “The results from multiple bioinformatics analyses indicate that hsa-miR-200c-3p, hsa-miR-885-5p, and hsa-miR-378f could be involved in the processes of pathological change in XELOX-chemotherapy-induced peripheral neuropathy.” – The results only show a significant correlation for hsa-miR-378f.

Minor Comments:

  1. 36: “Gastric cancer (GC) is one of the most serious health problems, being the fifth most common malignancy globally, and remains the third most common cause of cancer death worldwide [1,2]. According to GLOBOCAN 2018 data, GC was estimated to have caused the death of 783,000 people worldwide [2,3]” – This information is outdated, please refer to Globocan 2020 (PMID: 33538338).
  2. 49-51: “They regulate the expression of their target messenger RNAs (mRNAs) and can pro mote tumor growth, infiltration, angiogenesis, and antigenic escape” - Although these are widely accepted mechanisms, it is on the readers best interest if the authors provide a reference for each example mentioned.
  3. 74: NCI-CTCAE – uncommon abbreviation. Please refer to the full name the first time it is mentioned. It is only described the second time it is mentioned.

Figure 1b: in this picture it is hard to read the name of the miRNAs. Please provide an image with a better resolution

L 145-151: Please indicate the GO identifiers when referring to them.

Table 4: It is clearer if you substitute Count for Gene Count in the label.

Section 4.3: This section lacks detail. It is impossible to reproduce these results without knowing which packages and tools were used to perform the analysis.

Line 309: GlueGo --> ClueGo

Author Response

Response to Reviewer 1 Comments

This study aimed at identifying miRNAs associated with neurological side effects in GC patients who were receiving XELOX treatment. The authors found that XELOX treatment causes alterations in the expression of hsa-miR-22 200c-3p, hsa-miR-885-5p, and hsa-miR-378f, according to the findings. However, they could only validate these results for hsa-miR-378f using qRT-PCR, which is also the only one with correlation to GC. The authors present the ethics approval and informed consent protocol. The methods are well described, besides the bioinformatic analysis and the authors clearly state the limitations of the study. I have some concerns with this study presented point by point below.

Author’s response: We deeply appreciate your time and careful review. In accordance with your comments, we have made the appropriate revisions to the manuscript.

Major comments

Point 1: If the authors could only validate one miRNA and this miRNA was also the one which correlated with GC, why do GO, KEGG and network analysis with targets of two other miRNAs that were not validate nor correlated with GC?

Suggestion: Redo the analysis with the predicted targets of hsa-miR-378f only.

Response 1: We appreciate the reviewer for insightful comment. We redid the analysis with the predicted targets of hsa-miR-378f only.

(Page 1, line 28-32/ page 5, line 140-141, Supplementary Table S1/ page 6, line 148-155, Figure 3/ page 7, Figure 4/ page 8, line 166-169, Figure 5, Table 4, Supplementary Table S2/ page 10, line 214, line 218-220/ page 15, line 419-420 in revised manuscript)

[ref] Xiang, H.C.; Lin, L.X.; Hu, X.F.; Zhu, H.; Li, H.P.; Zhang, R.Y.; Hu, L.; Liu, W.T.; Zhao, Y.L.; Shu, Y. AMPK activation attenuates inflammatory pain through inhibiting NF-κB activation and IL-1β expression. J. Neuroinflammation 2019, 16, 34.

Point 2: 89: “using a volcano plot and heatmap.” Please refer to the methods used and not their graphic representation.

Response 2: In accordance with the reviewer’s comments, we referred to the methods used. In addition, we've added explanations for Figures 1a and 1b, and included the results for each method.

(Page 3, line 91-95, line 99-102 in revised manuscript)

Point 3: Section 4.3: This section lacks detail. It is impossible to reproduce these results without knowing which packages and tools were used to perform the analysis.

Response 3: In accordance with the reviewer’s comments, we added details for Section 4.3. First, we mentioned the NextSeq500 system, which an equipment for small RNA sequencing. Additionally, we indicated tool for miRNA target prediction and miRNA and target analysis. Finally, we revised the description of validation.

(Page 11, line 245-247, line 250-257 in revised manuscript)

Point 4: L-323-326: “The results from multiple bioinformatics analyses indicate that hsa-miR-200c-3p, hsa-miR-885-5p, and hsa-miR-378f could be involved in the processes of pathological change in XELOX-chemotherapy-induced peripheral neuropathy.” – The results only show a significant correlation for hsa-miR-378f.

Response 4: In accordance with the reviewer’s comments, we revised the results observed in this study in conclusions.

(Page 13, line 326-328 in revised manuscript)

Minor Comments

Point 5: 36: “Gastric cancer (GC) is one of the most serious health problems, being the fifth most common malignancy globally, and remains the third most common cause of cancer death worldwide [1,2]. According to GLOBOCAN 2018 data, GC was estimated to have caused the death of 783,000 people worldwide [2,3]” – This information is outdated, please refer to Globocan 2020 (PMID: 33538338).

Response 5: We deeply apricate reviewer’s kind and detail comments. We referred to Globocan 2020 data.

(Page 1, line 40-41/ page 14, line 351-353 in revised manuscript)

[ref] Sung, H.; Ferlay, J.; Siegel, R.L.; Laversanne, M.; Soerjomataram, I.; Jemal, A.; Bray, F. Global Cancer Statistics 2020: GLO-BOCAN Estimates of Incidence and Mortality Worldwide for 36 Cancers in 185 Countries. CA Cancer J. Clin. 2021, 71, 209-249.

[ref] Ilic, M.; Ilic, I. Epidemiology of stomach cancer. World J. Gastroenterol. 2022, 28, 1187-1203.

Point 6: 49-51: “They regulate the expression of their target messenger RNAs (mRNAs) and can pro mote tumor growth, infiltration, angiogenesis, and antigenic escape” - Although these are widely accepted mechanisms, it is on the readers best interest if the authors provide a reference for each example mentioned.

Response 6: In accordance with the reviewer’s comments, we cited the references for each example mentioned.

(Page 2, line 53/ page 14, line 364-371 in revised manuscript)

[ref] Stahlhut, C.; Slack, F.J. MicroRNAs and the cancer phenotype: profiling, signatures and clinical implications. Genome Med. 2013, 5, 111.

[ref] Suzuki, H.I.; Katsura, A.; Matsuyama, H.; Miyazono, K. MicroRNA regulons in tumor microenvironment. Oncogene 2015, 34, 3085-3094.

[ref] Tang, Y.; Zong, S.; Zeng, H.; Ruan, X.; Yao, L.; Han, S.; Hou, F. MicroRNAs and angiogenesis: a new era for the management of colorectal cancer. Cancer Cell Int. 2021, 21, 221.

[ref] Yang, Q.; Cao, W.; Wang, Z.; Zhang, B.; Liu, J. Regulation of cancer immune escape: The roles of miRNAs in immune checkpoint proteins. Cancer Lett. 2018, 431, 73-84.

Point 7: 74: NCI-CTCAE – uncommon abbreviation. Please refer to the full name the first time it is mentioned. It is only described the second time it is mentioned.

Response 7: In accordance with the reviewer’s comments, we referred to the full name the NCI-CTCAE.

(Page 2, line 75-77 in revised manuscript)

Point 8: Figure 1b: in this picture it is hard to read the name of the miRNAs. Please provide an image with a better resolution.

Response 8: In accordance with the reviewer’s comments, we provided an image with a better resolution for easy to read the name of miRNAs.

(Page 4, Figure 1 in revised manuscript)

Point 9: L 145-151: Please indicate the GO identifiers when referring to them.

Response 9: In accordance with the reviewer’s comments, we indicated the GO identifiers.

(Page 6, line 148-155 in revised manuscript)

Point 10: Table 4: It is clearer if you substitute Count for Gene Count in the label.

Response 10: In accordance with the reviewer’s comments, we substituted Gene Count for Count in the Table 4.

(Page 8, Table 4 in revised manuscript)

Point 11: Section 4.3: This section lacks detail. It is impossible to reproduce these results without knowing which packages and tools were used to perform the analysis.

Response 11: In accordance with the reviewer’s comments, we added details for Section 4.3. First, we mentioned the NextSeq500 system, which an equipment for small RNA sequencing. Additionally, we indicated tool for miRNA target prediction and miRNA and target analysis. Finally, we revised the description of validation.

(Page 11, line 245-247, line 250-257 in revised manuscript)

Point 12: Line 309: GlueGo --> ClueGo

Response 12: We apologize for our error. In accordance with the reviewer’s comments, we’ve corrected the typo.

(Page 13, line 313 in revised manuscript)

Author’s response: We would like to thank the reviewer again for taking the time to review our manuscript.

Reviewer 2 Report

The manuscript submitted by Yeongdon Ju focuses on a miRNA-seq performed on GC patients subjected to the XELOX regimen. The authors determined that among the plethora of dysregulated miRNAs, the hsa-miR-378f is significantly downregulated in those patients treated with more than 4 cycles of chemotherapy indicating the possibility to be associated with oxaliplatin-induced neuropathy. The work was carried out with adequate methods and even though the molecular mechanisms were not investigated, it is undoubted that miR-378f might play a major role in the side-effect induced by oxaliplatin therapy.

I strongly encourage the authors to discuss in the relative section that the origin of such a miRNA is unpredictable with the current methods of analysis, in particular, that miR-378f, analyzed in the whole plasma, might be generated by tumor or healthy cells in response to the oxaliplatin-induced oxidative stress and that it may be in a free form, for example in case it is released by apoptotic/necrotic cells, or loaded into extracellular vesicles in a sort of intracellular communication. Moreover, it could be interesting, for future studies, in case miR-378f is secreted by tumor cells, to determine if XELOX therapy strikes oxaliplatin-sensitive cells, promoting the downregulation of the analyzed miRNA or conversely if the putative resistant cells downregulate the miRNA to resist to therapy. An RNA-Scope or an FFPE-qPCR on tumor biopsies might partially explain the origin of this phenomenon but this would only be a suggestion and not a request.

Indeed, I do believe that the manuscript is publishable in its present form.

Minor request: In row 74 the words “both groups” were repeated.

Author Response

Response to Reviewer 2 Comments

The manuscript submitted by Yeongdon Ju focuses on a miRNA-seq performed on GC patients subjected to the XELOX regimen. The authors determined that among the plethora of dysregulated miRNAs, the hsa-miR-378f is significantly downregulated in those patients treated with more than 4 cycles of chemotherapy indicating the possibility to be associated with oxaliplatin-induced neuropathy. The work was carried out with adequate methods and even though the molecular mechanisms were not investigated, it is undoubted that miR-378f might play a major role in the side-effect induced by oxaliplatin therapy.

I strongly encourage the authors to discuss in the relative section that the origin of such a miRNA is unpredictable with the current methods of analysis, in particular, that miR-378f, analyzed in the whole plasma, might be generated by tumor or healthy cells in response to the oxaliplatin-induced oxidative stress and that it may be in a free form, for example in case it is released by apoptotic/necrotic cells, or loaded into extracellular vesicles in a sort of intracellular communication. Moreover, it could be interesting, for future studies, in case miR-378f is secreted by tumor cells, to determine if XELOX therapy strikes oxaliplatin-sensitive cells, promoting the downregulation of the analyzed miRNA or conversely if the putative resistant cells downregulate the miRNA to resist to therapy. An RNA-Scope or an FFPE-qPCR on tumor biopsies might partially explain the origin of this phenomenon but this would only be a suggestion and not a request.

Indeed, I do believe that the manuscript is publishable in its present form.

Author’s response: We deeply appreciate your time and careful review.

Minor request

Point 1: In row 74 the words “both groups” were repeated.

Response 1: In accordance with the reviewer’s request, we revised the repeat word ‘both groups’ to a single word.

(Page 2, line 77 in revised manuscript)

Author’s response: We would like to thank the reviewer again for taking the time to review our manuscript.

Round 2

Reviewer 1 Report

The authors have addressed every issue I raised, improving the manuscript quality, clarity and rational.